# Physicochemical and Stability Evaluation of Topical Niosomal Encapsulating Fosinopril/γ-Cyclodextrin Complex for Ocular Delivery

**DOI:** 10.3390/pharmaceutics14061147

**Published:** 2022-05-27

**Authors:** Hay Marn Hnin, Einar Stefánsson, Thorsteinn Loftsson, Rathapon Asasutjarit, Dusadee Charnvanich, Phatsawee Jansook

**Affiliations:** 1Faculty of Pharmaceutical Sciences, Chulalongkorn University, 254 Phyathai Road, Pathumwan, Bangkok 10330, Thailand; haymarn793@gmail.com (H.M.H.); dusadee.v@chula.ac.th (D.C.); 2Department of Ophthalmology, Faculty of Medicine, National University Hospital, University of Iceland, Landspitalinn, IS-101 Reykjavik, Iceland; einarste@landspitali.is; 3Faculty of Pharmaceutical Sciences, University of Iceland, Hofsvallagata 53, IS-107 Reykjavik, Iceland; thorstlo@hi.is; 4Faculty of Pharmacy, Thammasat University, 99 Moo 18 Paholyothin Road, Klong Luang, Rangsit 12120, Thailand; rathapon@tu.ac.th

**Keywords:** cyclodextrin, ophthalmic, fosinopril sodium, niosomes, encapsulation, stabilization

## Abstract

This study aimed to develop a chemically stable niosomal eye drop containing fosinopril (FOS) for lowering intraocular pressure. The effects of cyclodextrin (CD), surfactant types and membrane stabilizer/charged inducers on physiochemical and chemical properties of niosome were evaluated. The pH value, average particle size, size distribution and zeta potentials were within the acceptable range. All niosomal formulations were shown to be slightly hypertonic with low viscosity. Span^®^ 60/dicetyl phosphate niosomes in the presence and absence of γCD were selected as the optimum formulations according to their high %entrapment efficiency and negative zeta potential values as well as controlled release profile. According to ex vivo permeation study, the obtained lowest flux and apparent permeability coefficient values confirmed that FOS/γCD complex was encapsulated within the inner aqueous core of niosome and could be able to protect FOS from its hydrolytic degradation. The in vitro cytotoxicity revealed that niosome entrapped FOS or FOS/γCD formulations were moderate irritation to the eyes. Furthermore, FOS-loaded niosomal preparations exhibited good physical and chemical stabilities especially of those in the presence of γCD, for at least three months under the storage condition of 2–8 °C.

## 1. Introduction

Glaucoma is a multifactorial long term ocular neuropathy, which is associated with a progressive loss of visual field, structural abnormalities of retinal nerve fiber and cupping of the optic nerve head [1,2]. Recently, it has become the second leading cause of blindness worldwide after cataracts [3]. It was estimated that the primary open angle glaucoma cases in adult population will be risen up to 79.76 million in 2040 [4]. Many predictors for glaucoma have been identified, including age, positive family history, race, myopia and exfoliation syndrome [5]. Currently, intraocular pressure (IOP) is a major known risk factor for glaucoma. To lower IOP, treatment options involve oral and topical medications, laser therapy and surgical operation. Effective drug therapies include the drugs that reduce the rate of aqueous humor production and/or enhance its drainage. Several classes of drugs are available in managing long-term treatment of glaucoma, such as prostaglandin analogues, carbonic anhydrase inhibitors, α-adrenergic agonists, β-adrenergic blockers, and cholinergic agonists [1,2].

Angiotensin-converting enzyme (ACE) inhibitors have recently received attention as a new class of drug possessing the ability to lower IOP to treat glaucoma [6,7,8]. ACE is responsible for the conversion of the biologically inactive angiotensin I to the potent vasopressor, angiotensin II as well as the breakdown of bradykinin. Inhibition of ACE leads to the accumulation of bradykinin and promote the synthesis of prostaglandins, which could in turn lower IOP by increasing the uveoscleral outflow [9]. They also have a beneficial effect on retarding the progression of diabetic retinopathy in type II diabetic patients [10,11]. Moreover, ACE inhibitors showed beneficial effect in age-related macular degeneration [12]. Of these, fosinopril (FOS), the ester prodrug of fosinoprilat, and the first orally active phosphorus-containing ACE inhibitor, is an interesting compound to be used for lowering IOP. However, hydrolysis degradation of FOS was found in all conditions, i.e., acidic, basic and neutral, whereas the greater extent in basic condition [13]. Our previous study reported that the application of γ-cyclodextrin (γCD) as an inclusion complex could be able to enhance the solubility and chemical stability of FOS in aqueous solution [14].

Recently, colloidal drug delivery has been introduced as an alternative formulation approach for problematic drug candidates. Numerous colloidal carriers such as liposomes, niosomes, nanoparticles, microemulsions and micelles have been developed, which are applicable not only to solving the problems of poor solubility and stability but also to providing specific drug targeting, optimizing drug release properties and reducing toxicity [15]. As a vesicular carrier, niosome has gained attention because of its advantages including: (i) enhanced solubility and permeability; (ii) improved chemical stability; (iii) simple and cost-effective fabrication and (iv) low toxicity and high compatibility because of their nonionic nature [16].

Niosomes are nonionic surfactant vesicles, rising from the self-assembly of nonionic amphiphiles in aqueous media. The spherical shaped niosomes are capable of entrapping lipophilic molecules within the lipid bilayer by interacting with alkyl chains of nonionic surfactants, whereas hydrophilic drug molecules are located within an aqueous core by interacting with polar head groups of nonionic surfactants [17,18]. Numerous studies have reported the successful use of niosomes as ocular drug delivery carriers [19,20,21,22,23]. Vesicular delivery systems used in ophthalmic applications offer targeting at the site of action, improving chemical stability of encapsulated drugs and providing controlled release action at the corneal surface [24,25]. Vyas et al. (1998) reported that the ocular bioavailability of niosome entrapped water-soluble drugs, i.e., timolol maleate, increased as compared with timolol maleate solution [19]. This can be explained in that surfactants behave as penetration enhancers by removing the mucus layer and breaking junctional complexes [26].

In this study, niosomal eye drop preparations containing FOS alone or FOS/γCD inclusion complex were developed. The combined strategies, i.e., CD inclusion complex incorporated into a niosomal vesicle was applied to increase the chemical stability and to provide controlled drug release action. The physicochemical and chemical properties of niosomal formulations were evaluated. In addition, in vitro release, ex vivo permeation, in vitro cytotoxicity, and physical and chemical stability studies were also determined.

## 2. Materials and Methods

### 2.1. Materials

Fosinopril sodium (FOS) was purchased from Dideu Industries Group, Ltd. (Shaanxi, China). γ-Cyclodextrin (γCD) was purchased from Cyclolab (Budapest, Hungary). Polyoxyethylene 10 stearyl ether (Brij**^®^** 76) was distributed by The East Asiatic Public Company Ltd., (Bangkok, Thailand). Sorbitan monostearate (Span**^®^** 60) and poly-24-oxyethylene cholesteryl ether (Solulan**^®^** C-24, SC24) were kindly donated by Chemico Inter Corporation Ltd**.** (Bangkok, Thailand). Cholesterol, dicetyl phosphate (DCP) and stearylamine (STA) were received from Sigma-Aldrich (St. Louis, MO, USA), ethylenediamine tetra-acetic acid disodium salt (EDTA) and sodium metabisulfite (Na-MS) from Ajax Finechem Pty Ltd**.** (Taren Point, Australia). Semi-permeable cellophane membranes (SpectaPor**^®^**, molecular weight cut-off (MWCO) 12–14,000 Da) were obtained from Spectrum Europe (Breda, The Netherlands). All other chemicals used were of analytical reagent grade purity. Milli-Q (Millipore, Billerica, MA, USA) water was used to prepare all solutions.

### 2.2. Preparation of Niosomal Formulations Containing FOS

Niosome was prepared using thin-film hydration method. The niosome formulations were composed of nonionic surfactant, cholesterol, and membrane stabilizer/charged inducer at the mole ratio of 47.5: 47.5: 5. This ratio was optimized and shown to possess relatively good physicochemical characteristics obtained from blank niosome preparations. The total lipid composition was prepared at 100 μM in 5 mL of hydration medium (10 mM phosphate-buffered saline (pH 7.4) containing 1% (*w/v*) FOS, 0.1% (*w/v*) EDTA and 0.1% (*w/v*) Na-MS). The surfactants used in this study included Span^®^ 60 and Brij^®^ 76. Nonionic SC24 was used as a steric stabilizer, while positively charged STA and negatively charged DCP were used to provide the electrostatic stabilization of vesicles. Briefly, accurately weighed amounts of nonionic surfactant, cholesterol and membrane stabilizer/charge inducer were dissolved in 10 mL of chloroform in a 1 L round-bottom flask. The lipid mixture was slowly evaporated under reduced pressure at 40 °C using a rotary evaporator (Rotavapor R-200, BÜCHI Labortechnik AG, Flawil, Switzerland) with a constant rotation speed. The flask was partially immersed in a water bath and evaporated until a dried thin film appeared on the inner wall of the flask. Then, the formulation was kept in a desiccator under vacuum for 2 h to ensure the total removal of trace solvents. After that, dried lipid film was hydrated with 5 mL of hydration medium with and without 5% (*w/v*) γCD. Our previous work reported that EDTA and Na-MS are powerful antioxidants to protect FOS degradation [14]. The hydration of dried film was carried out by rotating the flask in a water bath at 60 °C for 30 min using a rotavapor under normal pressure. The size reduction was made by sonicating in an ultrasonic bath (GT sonic, GT SONIC Technology Park, Guangdong, China) at 60 °C for 30 min. To complete annealing and partition of the drug between the lipid bilayer and the aqueous phase, the formulation was left overnight at room temperature and then stored at 4 °C until subjected to analysis. The compositions of niosome formulae are shown in Table 1.

### 2.3. Physicochemical and Chemical Characterizations

#### 2.3.1. Osmolality, pH and Viscosity Determination

The pH values of all formulations were measured using a pH meter (SevenCompact S220-Micro, Mettler Toledo, Gießen, Germany) at 25 °C. The viscosity was determined by viscometer (Sine-wave Vibro SV-10, A&D Company, Limited, Tokyo, Japan) using the tuning-fork vibration method with frequency of 30 Hz at 25 °C and 34 °C. The osmolality was determined by osmometer (OSMOMAT 3000 basic, Gonotec GmbH, Berlin, Germany) at room temperature using the freezing point depression principle. All measurements were determined in triplicate.

#### 2.3.2. Particle Size, Size Distribution, and Zeta Potential

The particle size, size distribution and zeta potential of FOS-loaded niosome formulations were measured using the dynamic light scattering (DLS) technique (Zetasizer ^TM^ Nano ZS with software, Version 7.11, Malvern, UK). The measurements were carried out at a scattering angle of 180° and a temperature of 25 °C, a medium viscosity of 0.8872 mPa.s and a medium refractive index of 1.330. The concentration of niosome preparation was 20 μM. The particle size distribution was expressed as polydispersity (PDI). The particle size, size distribution and zeta potential were automatically calculated and analyzed using the software included within the system. Each measurement was performed in triplicate.

#### 2.3.3. Determining Drug Content and Entrapment Efficiency (EE)

The FOS was quantitatively determined using a reversed-phase HPLC component system from Agilent 1260 Infinity II consisting of a liquid chromatography pump (quaternary pump, G7111A), diode array UV-Vis detector (DAD, G7115A), auto sampler (G7129A) with Chem Station Software, Version E.02.02 and Phenomenex Kinetex 5 µm C18 reverse-phase column (150 × 4.6 mm) with C18 guard cartridge column MG II 5 μm, 4 × 10 mm. The HPLC conditions were as described below. The mobile phase comprised aqueous solution containing 1% (*v/v*) tetrahydrofuran and 0.05% (*v/v*) phosphoric acid: acetonitrile (30:70 volume ratio); a flow rate of 0.9 mL/min; wavelength of 205 nm; injection volume of 20 μL; column oven temperature of 40 °C; and run time of 6 min. The analytical method validation was performed to satisfy the validation criteria.

Total FOS content in niosomal preparation was determined by dissolving 100 µL of the sample in 10 mL of methanol:water (50:50 *v/v*). After proper dilution, the solution was filtered through a 0.45 μm nylon filter and analyzed using HPLC. To determine the percentage of EE (%EE), the sample was ultra-centrifuged (CP100NX, Hitachi Koki Co., Ltd., Tokyo, Japan) at 18,000 rpm at 4 °C for 1 h. Then, the content of unentrapped drug in the supernatant was diluted with methanol: water (50:50 *v/v*) and quantified by HPLC. All samples were performed in triplicate. The %EE was calculated as Equation (1):(1)$$\%EE=\frac{\left(D_{t}-D_{s}\right)}{D_{t}}\times 100$$
where *D_t_* is the total FOS content and *D_s_* is the FOS content in the supernatant.

#### 2.3.4. Transmission Electron Microscopy (TEM) Analysis

The morphologic examinations of selected FOS-loaded niosomes with or without γCD were performed using the TEM technique. Initially, the sample was placed on a formvar-coated grid. After blotting the grid with a filter paper, the grid was transferred onto a drop of negative stain. Aqueous 1% phosphotungstic acid solution was used as a negative stain. The sample was air dried at room temperature and finally the samples were examined by TEM (Model JEM-2100F, JEOL, Peabody, MA, USA).

### 2.4. In Vitro Release Study

The in vitro release study was performed using a modified Franz diffusion cell apparatus consisting of donor and receptor chambers (NK Laboratories Co., Ltd., Bangkok, Thailand). These two chambers were separated by a semipermeable membrane (MWCO 12,000–14,000 Da). The membrane was presoaked overnight in the receptor phase consisting of phosphate-buffered saline (PBS, pH 7.4). The receptor phase was degassed to remove dissolved air before being placed in the receptor chamber. The sample (1.5 mL) of each niosomal formulation was placed in the donor chamber. The receptor phase was continuously stirred at 150 rpm throughout the experiment and a controlled temperature was maintained at 34 ± 1 °C by a thermostated circulating bath (GRANT W6, Akribis Scientific Limited, Cheshire, UK). A 150 μL aliquot of the receptor medium was withdrawn at timed intervals and replaced immediately with an equal volume of fresh receptor phase. The FOS content in the receptor medium was determined using HPLC and the amount of cumulative drug release was calculated. Each formulation was performed in triplicate.

### 2.5. Ex Vivo Permeation Study

The ex vivo permeation study was performed across the cornea and sclera of porcine eyes obtained within 4 h after the death of pigs from a slaughterhouse. In this study, the cornea and sclera were dissected from porcine eyes and replaced with the semipermeable cellophane membrane as previously described in in vitro release study. The selected FOS-loaded Span**^®^** 60-niosomal formulations and an aqueous saturated solution of FOS/γCD complex used as a control were conducted at least in triplicate. The FOS content in the receptor phase at timed intervals was determined using HPLC. The steady state flux was calculated as the slope of linear section of the amount of drug in the receptor chamber (*q*) versus time (*t*) profiles, and the apparent permeability coefficient (*P*_app_) was calculated from the flux (*J*) according to Equation (2):(2)$$J=\frac{d_{q}}{A\;\cdot \;d_{t}}=P_{app}\;\cdot \;C_{d}$$
where *A* is the surface area of the mounted membrane (1.7 cm^2^) and *C_d_* is the initial concentration of the drug in the donor chamber. 

### 2.6. Cell Viability and Short Time Exposure (STE) Test

In vitro cytotoxicity test was determined using the methylthiazolyl-diphenyl-tetrazolium bromide (MTT) assay [27,28]. Briefly, the niosomal formulations containing FOS without and with γCD (Sp-DCP and Sp-DCP+γCD, respectively) including their respective blank samples, i.e., B-Sp-DCP and B-Sp-DCP+γCD were evaluated for their toxicity to the rabbit corneal fibroblasts, i.e., the SIRC (rabbit corneal cell line) cells (CCL-60; ATCC, Manassas, VA, USA). Each sample was diluted to the concentration of 0.5, 1, 2, 5 and 10% (*v/v*) of the test samples by a complete medium that contained Eagle’s Minimum Essential Medium and fetal bovine serum (FBS). FOS concentrations in the tested samples ranged from 0.005 to 0.1% *w/v*. The cells were cultured in the complete medium and maintained at 37 °C under 5% CO_2_ atmosphere. They were seeded in 96-well plates with a density of 1 × 10^5^ cells/well/100 µL and incubated for 24 h. Thereafter, each test sample (100 µL) was added to the well. The cells were incubated for 24 h and washed twice with PBS (pH 7.4) at the end of incubation period. MTT solution in PBS (pH 7.4) was added to each well and incubated for 4 h. The formazan crystals were dissolved using 0.04 M HCl in isopropanol (100 µL/well). The optical density (OD) of each well was measured at 570 nm by a microplate reader (Fluostar Omega, BMG Labtech, Ortenberg, Germany). The experiments were performed in four replications, and cell viability (CV) was calculated following Equation (3). The test samples were considered to be toxic to the cells if the CV (%) was less than 70%.
(3)$$CV\left(\%\right)=\frac{OD_{sample}}{OD_{control}}\times 100$$
where the *OD_sample_* and *OD_control_* are an *OD* of the media from the wells containing the SIRC cells incubated with the samples and MTT solution, and an *OD* of media from the wells containing the cells incubated with MTT solution without the samples, respectively.

The eye irritation potential of those test samples was further evaluated based on the MTT reduction assay [29]. The in vitro eye irritation test was performed according to the procedure of the STE test proposed by Takahashi et al. (2008) [30]. The CV of SIRC cells was determined after they were exposed to 200 µL of either 5% or 0.05% of the test samples dispersed in normal saline for 5 min. The eye irritation potential from the STE test was scored following the criteria for STE irritation scoring. Then, the obtained scores from the 5% and the 0.05% tests were summed up to rank the eye irritation potential. The total scores were ranked as 1, 2 and 3, defined as minimal ocular irritant, moderate ocular irritant, and severe ocular irritant, respectively.

### 2.7. Physical and Chemical Stability Studies

To investigate the effect of γCD on stability of FOS in niosomal vesicles, selected optimal FOS-loaded niosomal formulations (in the presence and absence of γCD) and aqueous solution of FOS/γCD complex (as a control) were evaluated using the ongoing stability program following International Conference on Harmonization (ICH) guidelines [31]. The samples were stored in tightly closed glass vials at 4 °C, long term condition (30 ± 2 °C, 75 ± 5% relative humidity (RH)) and accelerated condition (40 ± 2 °C, 75 ± 5% RH). Physical appearance was assessed, and formulations were analyzed with respect to pH, particle size and size distribution, zeta potential and the FOS content at timed intervals of 0, 1, 3 and 6 months.

### 2.8. Statistical Analysis

All quantitative data were presented as means ± standard deviation (SD). The data were statistically calculated using one-way ANOVA (SPSS Software, Version 16.0, SPSS Inc., Chicago, IL, USA). The *p* < 0.05 was considered statistically significant.

## 3. Results and Discussion

### 3.1. Physicochemical and Chemical Characterizations of Niosomal Formulations Containing FOS

#### 3.1.1. Osmolality, pH and Viscosity

Table 2 shows the osmolality, pH and viscosity values of FOS-loaded niosomal formulations. The pH values of all formulations were in the range of 6.7 to 7.2, which was acceptable and very close to the ideal pH for the eye drop, i.e., 7.2 ± 0.2 [32]. The slightly lower pH values were found by adding γCD but without significance (*p* > 0.05). All niosomal preparations were at a low viscosity of about 1 to 2 mPa.s. The low viscosity preparation is expected to easily spread on the eye surface and not affect the vision, and it is unlikely to cause any lacrimation or blurredness [33]. Conversely, due to the absence of viscosity-inducing agents, instillation of eye drops may be required several times a day. As expected, the viscosity measured at 34 °C was slightly lower than that measured at 25 °C [34]. All formulations were slightly hypertonic and beyond the acceptable values (within 260 to 330 mOsm/kg). Due to the osmotic property of CDs, osmolality was found to be higher in preparations containing γCD. However, hypertonic eye drops are better tolerated than hypotonic eye drops and they also provide short term discomfort due to dilution with lachrymal fluid taking place rapidly after administration [35].

#### 3.1.2. Particle Size, Size Distribution and Zeta Potential

The particle size and size distribution of FOS-loaded niosomal formulations measured by DLS technique are shown in Table 3. The average particle size was found to range from 190 to 270 nm, and PDI values were found between 0.1 and 0.5. This demonstrated polydisperse sample with heterogenous population of particles. In lipid-based nanoparticles, a PDI value of 0.3 and below indicates a homogenous population and is considered to be an acceptable nanocarrier for drug delivery systems [36]. Thus, further steps in the manufacturing process, such as extrusion or high-pressure homogenization, may be necessary to lower the PDI values for monodispersed systems. In most cases, the size of niosomes with Span^®^ 60 (HLB 4.7) were larger than those of Brij^®^ 76 (HLB 12.4). Vesicle size is generally known to be directly dependent on HLB value of the surfactant used where higher HLB produces larger size vesicles [37,38,39,40]. However, several studies have inversely reported that lower HLB values produce larger size vesicles [22,41,42]. This discrepancy is probably due to differing preparation methods, differing physiochemical properties of loaded drugs and the effect of membrane additives.

The addition of a membrane charge was observed to influence particle size (Table 3). Incorporating DCP in Span^®^ 60-niosome, i.e., Sp-DCP, produced relatively larger average particle sizes than those of STA followed by SC24 (Sp-STA and Sp-SC24, respectively). This could be explained by the similar charge of DCP, Span^®^ 60 and cholesterol head groups producing electrostatic repulsion among them, decreasing membrane curvature; and therefore, increasing particle size [43]. In contrast, in the case of Brij^®^ 76, vesicle size was found in the trend of SC24 > STA > DCP. This might be due to differences in the accommodating ability of surfactants among the membrane additives. Incorporating SC24 in hydrophilic Brij^®^ 76 surfactant led to increased membrane permeability and interstitial spaces between the bilayer membranes due to its bulky structures with long and highly hydrophilic poly-24-oxyethylene chains, resulting in increased in size [44].

Compared with the formulations with or without γCD, the preparations containing γCD displayed smaller mean particle size than those of the corresponding pure FOS-loaded niosomes. CDs form hydrophobic interactions with a hydrophobic tail as well as hydrogen bonding with the polar head group of nonionic surfactants [45]. Therefore, the complexation of CD with hydrophobic tails of surfactants resulted in lower packing density of incorporated surfactant and thereby decreased membrane thickness [46]. Additionally, the adsorption of γCD on surface modified niosomes also decreased vesicle size. This was due to CD interacting with polar head groups of surfactants through hydrogen bonding, leading to increased area of the polar head groups at the interphase as well as altering the radius of the curvature [47].

All FOS niosomal formulations exhibited negative zeta-potential values (Table 3). This might have been due to free hydroxyl groups present in cholesterol and surfactant molecules [48]. Because of the contribution of a negative charge due to ionization of the acidic (–HPO_4_) group by DCP, it produced a higher negative zeta potential value. The resultant electrostatic repulsion was likely to account for reducing the tendency of niosome aggregation. Conversely, STA introduced a positive charge via the protonation of the basic-NH_2_ group which adsorbed on the surface of niosome and exhibited lower negative zeta potential values through charge neutralization than the uncharged one, i.e., SC24 [49]. SC24 has no net charge and does not provide additional ions in dispersion media. It enhances membrane physical stability by providing steric stabilization [17]. It has been concluded that the highest negative zeta potential obtained by adding DCP could be of great importance to increase the stability and restraining niosomal dispersions from coalescence and aggregation during storage. Regarding niosomal formulations in the presence of γCD, lower zeta potential values were observed than those observed for corresponding nonCD-based niosomes. This was due to CD acting as a shell on the surface charge of niosome by hydrogen bond formation between the hydrophilic head group of surfactants with hydroxy groups on the exterior of CD [46,47,50,51].

#### 3.1.3. %EE of FOS-Loaded Niosomal Formulations

Lipophilic drugs are well known to be preferentially taken up by niosome compared with hydrophilic ones due to higher partitioning through the lipid phase of the vesicles [52]. The %EE values of 21 to 35% were obtained in Span^®^ 60-niosomes, which were relatively superior to those prepared with Brij^®^ 76 (Table 3). This might have been due to the lower HLB value of Span^®^ 60 (HLB 4.7) in contrast to Brij^®^ 76 (HLB 12.4). In addition, Span^®^ 60 has a higher transition temperature (Tc), i.e., 53 °C, compared with Brij^®^ 76 (34 °C) [53]. The surfactant with higher Tc usually forms less leaky vesicles; and thus, results in higher drug entrapment of water-soluble solutes [23,38].

The effect of stabilizer on %EE was found in the trend of DCP > SC24 > STA. The presence of double hydrocarbon chains in DCP imparted a greater packing of the bilayer membrane resulting in higher %EE. Due to the presence of highly hydrophilic poly-24-oxyethylene chains of SC24, the membrane becomes more flexible and permeable; thus decreasing %EE [44]. The lowest %EE by STA could be explained by an electrostatic induced chain tilt which subsequently changes the lateral packing of the bilayers [54]. This result was similar to the observation of the rupture of vesicles by the aggregation and fusion of vesicles under the polarized light microscope (data not shown).

According to our knowledge base, few studies have reported CD inclusion complex in niosome vesicles [47,55,56,57,58,59,60]. Our data results have shown that %EE of FOS increased when incorporating the FOS/γCD inclusion complex in niosomal preparations. This finding was similar to related reports [58,61]. The higher %EE in niosome containing γCD might have been because CD forms hydrogen bonds interacting with the polar head group of nonionic surfactants. The stronger the hydrogen binding intensity, the greater %EE was obtained [45,62]. Moreover, complexation of free CD with hydrophobic tails of surfactants creates a more internal aqueous space by decreasing membrane thickness [46,47]. However, all niosomal formulations have poor %EE of FOS (<40%). Remote loading method and changes to the formulation variables (i.e., surfactant/cholesterol ratio and their concentrations, buffer molarity and pH, hydration time, etc.) can be applied to optimize %EE. Due to the lower %EE of Brij^®^ 76-niosomes (stabilized by SC24 and DCP) and the evidence of the particle aggregation with the lowest %EE among the groups in all niosomes using STA as stabilizer, these formulations were excluded from further studies.

#### 3.1.4. TEM Analysis

The TEM micrographs of FOS-loaded Span^®^ 60-niosomes are shown in Figure 1. It demonstrated that the vesicles were well identified and presented in a nearly spherical shape. TEM images of niosomal formulations in the presence of γCD showed smaller particle size which corresponded to those determined by DLS measurement (Table 3). It has been observed that the small white spots distributed in niosome were stabilized by SC24 in the presence and absence of γCD (Figure 1a,c). Interestingly, in the case of DCP in the presence of γCD (Sp-DCP+γCD), the larger internal aqueous core was detected (Figure 1d) when compared with the one without γCD (Sp-DCP) (Figure 1b). The wider the hydrophilic core of niosome, the more capacity it could accommodate, including both hydrophilic drugs and water-soluble drug/CD complexes. Therefore, TEM micrographs showed a good correlation with the higher %EE of FOS-loaded Span^®^ 60/DCP niosome containing γCD (Table 3).

### 3.2. In Vitro Release Study 

The in vitro release profiles of selected FOS-loaded Span^®^ 60-niosomes are shown in Figure 2. Notably, a more controlled release manner was obtained from FOS-loaded niosomes stabilized by DCP than that obtained from those stabilized by SC24. Due to the parallel alignment of double hydrocarbon chains of DCP to the hydrocarbon chains of Span^®^ 60 as well as its parallel orientation of polar phosphate groups to the polar heads of Span^®^ 60, DCP provided more packing and filling in of any irregularities through the bilayer membrane. Such enhancement in the packing properties could render less membrane permeability to the entrapped water-soluble molecules and retard the drug release [44]. In both cases, FOS/γCD complexes that were entrapped niosomal formulations showed slower release rates than those of only FOS-loaded niosomes. Similar results have been reported with methotrexate where niosome with drug/βCD inclusion complexes produced relatively slower release pattern of the entrapped drug compared with both free drug incorporated niosome and drug/CD complex preparation [61]. Sheena et al. (1997) compared the release profiles of pilocarpine/βCD loaded and nonCD-based niosomal preparations. The result revealed that βCD-based niosomal formulations showed slower and more sustained release than that of conventional niosomes [58].

An important issue in evaluating reduced IOP among patients with glaucoma is 24 h control [63]. The more controlled release pattern of FOS niosomal preparation provides greater benefit for targeted glaucoma treatment. In contrast, the slow drug release may affect the insufficient therapeutic drug level in the ocular tissues. Niosomes have been investigated to enhance the poorly absorbed drug molecules by binding to the corneal surface and improving the contact time, thereby increasing the ocular bioavailability of drugs. To evaluate the FOS permeation through the ocular membranes, the optimum formulations, i.e., Sp-DCP and Sp-DCP+γCD were selected for further ex vivo permeation and stability studies.

### 3.3. Ex Vivo Permeation Study

The flux and *P*_app_ values of FOS-loaded Span^®^/DCP niosomal preparations in the presence and absence of γCD including aqueous solution containing FOS/γCD complex are displayed in Table 4. Notably, *P*_app_ level through sclera was higher than that of the cornea in all tested preparations. This might be due to the loose structural matrix and less complicated tissue layer of sclera [64,65]. According to the literature, the permeability of sclera is approximately 10 times greater than that of the cornea [66]. Thus, the scleral route is an alternative pathway to deliver drugs in both anterior and posterior segments of the eye. Loch et al. (2012) showed that the *P*_app_ values of ciprofloxacin, timolol and lidocaine for sclera are higher than those for the cornea [67]. Ahmed and Patton (1985) also revealed that intraocular penetration of a large molecule weight, i.e., insulin across the sclera was higher than those through the cornea [68].

In both cases of cornea and sclera, the flux and *P*_app_ values of FOS from niosomal preparations were significantly lower than those for the FOS/γCD complex preparation (*p* < 0.05) (Table 4). As expected, the FOS-loaded niosomes exhibited a more controlled drug release manner than that of the FOS/γCD complex preparation because the free drug or drug/CD inclusion complex had to be diffused from the inner aqueous core of the niosome through the lipid bilayer and then permeated through the membrane [69]. It has been supported and confirmed that FOS molecules in both free and inclusion complex forms were deposited in the inner core of niosomes. Regarding the effect of CD incorporated in niosomal formulations, both flux and *P*_app_ of niosome containing γCD (Sp-DCP+γCD) were lower than those without γCD (Sp-DCP). Again, it has been emphasized that most FOS molecules were included in the γCD cavity as inclusion complexes and were localized in the inner core of the niosome, i.e., high %EE. In addition, CD forms a strong hydrogen bonding interaction with the polar head group of nonionic surfactants, resulting in lower flux and *P*_app_ values of FOS-loaded niosome containing γCD.

### 3.4. Cell Viability and STE Test

Figure 3 shows the viability (%) of SIRC cells against the concentrations of unloaded and FOS-loaded Span^®^ 60/DCP niosomal formulations. As expected, the test samples at high concentrations were toxic to the SIRC cells. However, unloaded FOS Sp-DCP niosomes (blank) were safer than other formulations because the cell viability of the SIRC cells was greater than 70% at the entire concentrations around 5 to 0.5% *v/v*. Further, the others were safe to the SIRC cells at a concentration around 0.5% *v/v* only.

The in vitro irritation test was further evaluated. The STE test could provide representative information to the animal testing that involves the Draize test in rabbits [30]. The %CV of SIRC cells after exposure to 5% and 0.05% concentrations of niosomal formulation with loaded and unloaded FOS for 5 min are shown in Table 5. Notably, the total scores of eye irritation potential of both niosomal formulations entrapped FOS or FOS/γCD and their respective blank formulations were equal to 2. Thus, these formulations were defined as a moderate ocular irritant. On the other hand, this result demonstrated that FOS-loaded niosomal preparations could be conditionally accepted as safe for ophthalmic use. These observations might be due to the hyperosmolar solutions of the eye drop preparations.

### 3.5. Physical and Chemical Stability Studies of FOS 

The pH, particle size, size distribution, zeta potential and percent drug content were used as the parameters to evaluate the stability of FOS in niosomal formulations. In this study, two selected formulations, i.e., FOS-loaded Span^®^ 60/DCP niosomal formulations in the presence and absence of γCD were evaluated, and the aqueous solution of FOS/γCD complex was used as a control. The physical stability, i.e., pH, mean particle size, size distribution and zeta potential, of FOS after storage at 4 °C, in long term and accelerated conditions at various time intervals is shown in Appendix A. 

In the case of the aqueous solution of the FOS/γCD complex, the pH value slightly decreased at 4 °C but more obviously at higher temperatures. The particle size was significantly increased at all storage conditions and PDI values were out of specification at 30 and 40 °C. The zeta potential values also decreased under all conditions and significantly decreased at storage condition at 40 °C. It was concluded that FOS in the complexing aqueous medium exhibited low physical stability, especially the particle size growth upon storing for six months.

After storing for six months at 4 °C, a slightly decreased pH was found in both niosomal formulations; however, at higher storage temperatures of 30 and 40 °C, significantly reduced pH was detected (*p* < 0.05). This might have been due to a progressive increase in the hydrolysis of fatty acid in niosome with increasing temperature [70]. Regarding vesicle sizes and size distribution, both FOS-loaded niosomes had no appreciable changes at 4 °C, indicating a good physical stability. As expected, larger differences in these parameters were observed at higher temperatures of 30 and 40 °C. The particle size was exponentially increased and the PDI values were out of specification at 30 and 40 °C (PDI > 0.7) over the six-month period. The aggregation or fusion of vesicles generally occurred as molecular mobility increased and transformed to larger ones [71,72]. While particle size and size distributions indicate stability for particle-based formulations, %EE is considered as a stability-indicating parameter for this study in direct comparison to its non-particulate counterparts. Decreasing zeta potential values were found in all storage conditions but more significantly at higher temperatures. This lower zeta potential directly correlated to lower electrostatic repulsion and as a result, aggregation or fusion of vesicles resulted in increased particle size.

According to the six-month chemical stability data (Table 6), the drug content was significantly decreased in the aqueous solution consisting of the FOS/γCD complex representing 51, 8 and 3% at 4, 30 and 40 °C, respectively. Notably, FOS could not withstand an aqueous solution containing γCD. On the other hand, the CD inclusion complex was insufficient to enhance the chemical stability of FOS. We have found that the niosomal preparations revealed greater chemical stability than nonvesicular preparations, i.e., aqueous solutions containing the FOS/γCD complex at all storage conditions. Regarding the effect of γCD on chemical stability of FOS in niosome, Sp-DCP+γCD niosome showed relatively greater stability than Sp-DCP niosome at all storage temperatures. Under the refrigerated condition of 4 °C, 92% of FOS remained in Sp-DCP+γCD niosome, whereas only 88% remained in Sp-DCP niosome after storing for six months. Incorporating γCD as FOS/γCD complex in niosome showed relatively more stability than in that without CD. 

From the overall data results, the proposed drawings of FOS-loaded niosomes are shown in Figure 4. Niosomal platform could protect chemically unstable drug molecule, FOS by entrapping its inner the aqueous core. Additionally, the effect of γCD inclusion complex formation is the predominant factor to provide higher %EE of FOS in niosomal formulations by preventing the drug degradation via hydrolysis and consequently enhances the chemical stability of FOS in aqueous solution. 

## 4. Conclusions

To enhance the chemical stability of FOS in aqueous solution, niosomal formulations were developed. The effects of CD, surfactant type and membrane stabilizer/charged inducers on physiochemical and chemical properties of niosome were characterized. The average particle size was detected within the nanometer range and PDI values were within an acceptable range. The slow permeation rate of FOS through excised porcine cornea and sclera was obtained in γCD-loaded Span^®^ 60/DCP niosomal formulation. The chemical stability of FOS in the formation of γCD inclusion complex could not withstand the aqueous solution. Niosomal preparations with moderate irritation could prevent FOS degradation and they exhibited physical and chemical stability for at least three months at 4 °C. The optimum formulation to enhance the chemical stability of FOS consisted of FOS/γCD complex loaded niosome. To increase the shelf-file of the FOS niosomal formulation, the conversion to lyophilized powder for reconstitution is considered for further studies. Our studies successfully investigated the preformulation and ophthalmic formulation development of FOS. However, to demonstrate a clinically viable formulation, the in vivo pharmacokinetic in rabbit eye was considered for future perspective studies.

## Figures and Tables

**Figure 1 pharmaceutics-14-01147-f001:**
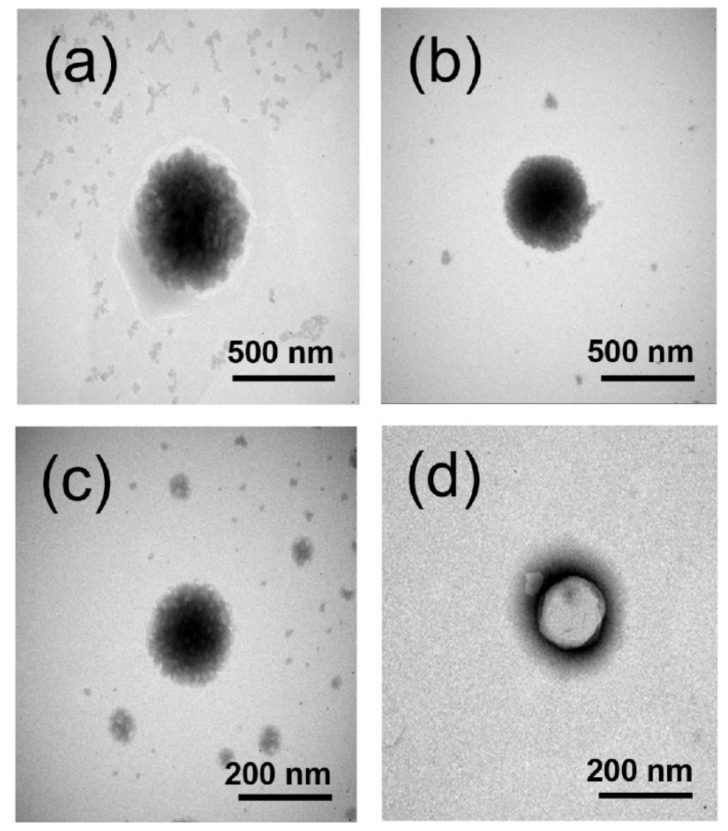
TEM micrographs of FOS-loaded Span^®^ 60-niosomes (**a**) Sp-SC24; (**b**) Sp-DCP; (**c**) Sp-SC24+γCD and (**d**) Sp-DCP+γCD.

**Figure 2 pharmaceutics-14-01147-f002:**
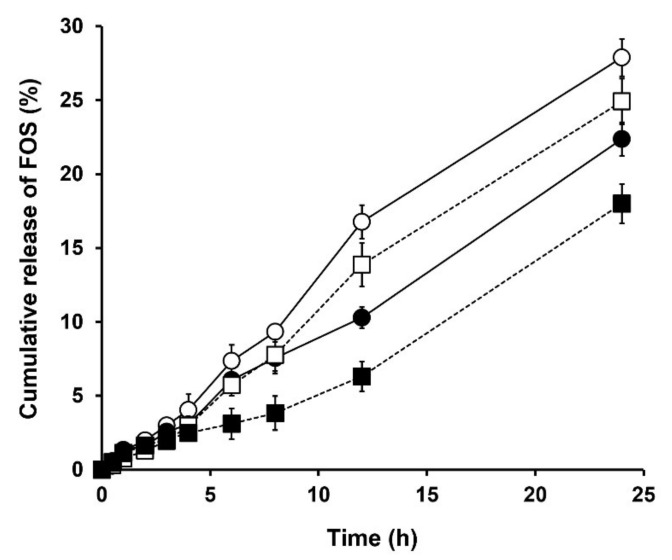
The release profiles of FOS-loaded Span^®^ 60-niosomes through semipermeable membrane with MWCO 12,000–14,000 Da; (○) Sp-SC24; (□) Sp-DCP; (●) Sp-SC24+γCD and (■) Sp-DCP+γCD.

**Figure 3 pharmaceutics-14-01147-f003:**
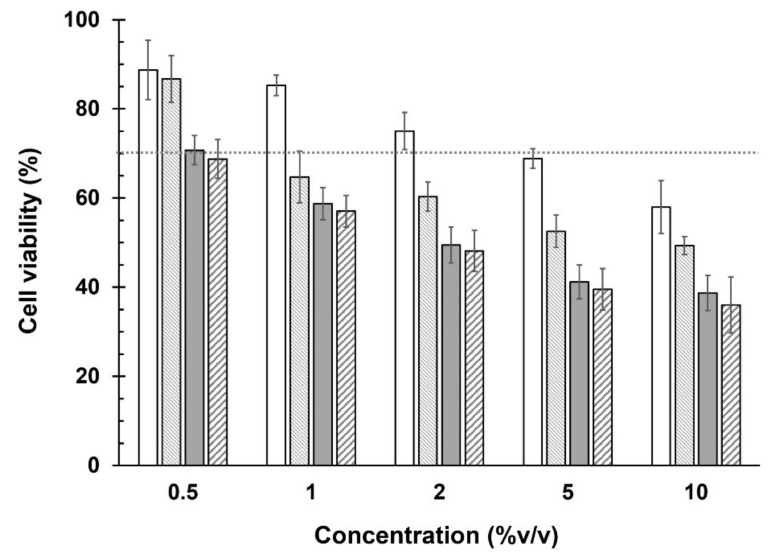
In vitro cytotoxicity test of FOS-loaded niosomal formulations, (
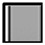
) Sp-DCP and (
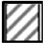
) Sp-DCP+γCD, and, (
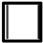
) blank Sp-DCP and (
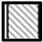
) blank Sp-DCP+γCD, at various concentrations in the SIRC cells (*n* = 4, mean ± SD).

**Figure 4 pharmaceutics-14-01147-f004:**
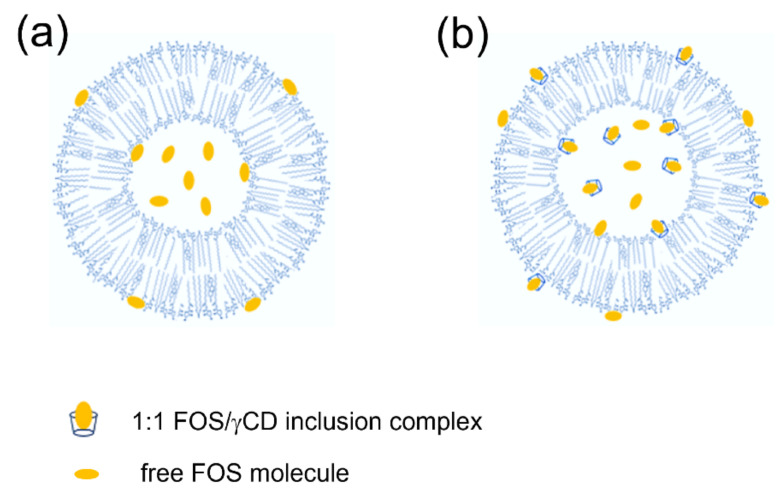
Proposed drawing of (**a**) FOS-loaded niosomes and (**b**) FOS/γCD loaded noisome.

**Table 1 pharmaceutics-14-01147-t001:** Compositions of FOS-loaded niosomal formulations.

Formulation ^a^	Span^®^ 60-Niosome						Brij^®^ 76-Niosome					
	Sp-SC24	Sp-DCP	Sp-STA	Sp-SC24+γCD	Sp-DCP+γCD	Sp-STA+γCD	Br-SC24	Br-DCP	Br-STA	Br-SC24+γCD	Br-DCP+γCD	Br-STA+γCD
*Ingredients in organic phase (μM)* ^b^												
Span^®^ 60	47.5	47.5	47.5	47.5	47.5	47.5	-	-	-	-	-	-
Brij^®^ 76	-	-	-	-	-	-	47.5	47.5	47.5	47.5	47.5	47.5
Cholesterol	47.5	47.5	47.5	47.5	47.5	47.5	47.5	47.5	47.5	47.5	47.5	47.5
SC24	5	-	-	5	-	-	5	-	-	5	-	-
DCP	-	5	-	-	5	-	-	5	-	-	5	^-^
STA	-	-	5	-	-	5	-	-	5	-	-	5
*Ingredients in aqueous phase (% w/v)* ^c^												
FOS	1	1	1	1	1	1	1	1	1	1	1	1
γCD	-	-	-	5	5	5	-	-	-	5	5	5

^a^ SC24, Solulan^®^C24; DCP, dicetylphosphate; STA, stearylamine; FOS, fosinopril sodium, ^b^ solubilized in 10 mL of chloroform, ^c^ solubilized in 5 mL of phosphate-buffered saline pH 7.4 containing 0.1% (*w/v*) EDTA and 0.1% (*w/v*) sodium metabisulfite.

**Table 2 pharmaceutics-14-01147-t002:** Osmolality, pH and viscosity values of the FOS-loaded niosomal formulations (*n* = 3, mean ± SD).

Formulation	pH	Osmolality (mOsm/kg)	Viscosity (mPa.s)	
			25 ± 1 °C	34 ± 1 °C
*Span^®^ 60-Niosome*				
Sp-SC24	7.02 ± 0.05	358 ± 5	1.48 ± 0.01	1.18 ± 0.01
Sp-DCP	6.73 ± 0.04	364 ± 6	1.81 ± 0.02	1.30 ± 0.01
Sp-STA	7.26 ± 0.03	366 ± 8	1.38 ± 0.02	1.12 ± 0.01
Sp-SC24+γCD	6.83 ± 0.03	372 ± 5	1.76 ± 0.02	1.50 ± 0.01
Sp-DCP+γCD	6.70 ± 0.03	374 ± 6	1.98 ± 0.02	1.72 ± 0.02
Sp-STA+γCD	6.75 ± 0.01	382 ± 5	1.75 ± 0.01	1.52 ± 0.01
*Brij^®^ 76-Niosome*				
Br-SC24	6.91 ± 0.01	346 ± 6	1.43 ± 0.01	1.21 ± 0.01
Br-DCP	6.95 ± 0.01	354 ± 8	1.64 ± 0.02	1.34 ± 0.01
Br-STA	7.22 ± 0.03	359 ± 10	1.41 ± 0.01	1.15 ± 0.01
Br-SC24+γCD	6.87 ± 0.02	364 ± 8	1.68 ± 0.02	1.34 ± 0.02
Br-DCP+γCD	6.78 ± 0.08	378 ± 3	1.86 ± 0.01	1.56 ± 0.01
Br-STA+γCD	6.86 ± 0.05	379 ± 9	1.65 ± 0.02	1.38 ± 0.01

**Table 3 pharmaceutics-14-01147-t003:** Mean particle size, size distribution, zeta potential and %EE of FOS-loaded niosomal formulations (*n* = 3, mean ± SD).

Formulation	Z-Average (d.nm)	Size Distribution (PDI)	Zeta Potential (mV)	%EE
Sp-SC24	245.1 ± 5.02	0.46 ± 0.03	−32.70 ± 1.64	21.34 ± 0.42
Sp-DCP	262.4 ± 5.00	0.45 ± 0.01	−37.70 ± 1.15	28.68 ± 0.77
Sp-STA	250.4 ± 6.31	0.35 ± 0.03	−15.43 ± 1.46	9.20 ± 0.30
Sp-SC24+γCD	198.0 ± 4.50	0.52 ± 0.01	−20.27 ± 0.67	25.99 ± 0.78
Sp-DCP+γCD	246.8 ± 3.71	0.42 ± 0.01	−27.17 ± 1.63	34.43 ± 0.80
Sp-STA+γCD	229.1 ± 5.16	0.36 ± 0.06	−13.40 ± 1.91	11.30 ± 0.85
Br-SC24	257.2 ± 4.29	0.32 ± 0.01	−24.30 ± 2.01	10.70 ± 0.27
Br-DCP	212.0 ± 0.72	0.36 ± 0.03	−34.97 ± 0.35	12.94 ± 0.57
Br-STA	214.8 ± 4.01	0.37 ± 0.02	−7.41 ± 0.40	7.73 ± 0.97
Br-SC24+γCD	246.0 ± 0.96	0.11 ± 0.02	−21.20 ± 1.04	12.58 ± 0.85
Br-DCP+γCD	200.0 ± 1.87	0.32 ± 0.01	−23.73 ± 1.97	14.02 ± 0.10
Br-STA+γCD	211.6 ± 1.52	0.34 ± 0.05	−6.94 ± 0.43	8.09 ± 0.80

**Table 4 pharmaceutics-14-01147-t004:** Flux and apparent permeation coefficient (*P*_app_) of FOS-loaded Span^®^ 60/DCP niosomal formulations in the presence and absence of γCD and aqueous FOS/γCD complex solution, through porcine cornea or sclera (*n* = 4, Mean ± SD).

Formulation	Cornea		Sclera	
	Flux ± S.D. (μgh^−1^ cm^−2^)	*P*_app_ ± S.D. (×10^−6^ cms^−1^)	Flux ± S.D. (μgh^−1^ cm^−2^)	*P*_app_ ± S.D. (×10^−6^ cms^−1^)
Sp-DCP	31.086 ± 6.32	0.920 ± 0.18	40.066 ± 40.35	1.155 ± 0.11
Sp-DCP+γCD	22.843 ± 7.95	0.635 ± 0.21	33.092 ± 2.38	0.927 ± 0.08
FOS/γCD complex	62.794 ± 6.23 ^a^	1.870 ± 0.18 ^a^	86.762 ± 5.25 ^a^	2.583 ± 0.16 ^a^

^a^ Statistically significant difference compared with FOS-loaded niosomal formulations (*p* < 0.05).

**Table 5 pharmaceutics-14-01147-t005:** Scores obtained from the short time exposure (STE) test of the test samples.

Concentration of the Test Samples	Test Samples	%CV of the SIRC Cells	Criteria for Scoring	Obtained Scores
(I) 5%	(1) Blank Sp-DCP	67 ± 5	If CV >70%: scored 0	1
	(2) Blank Sp-DCP + γCD	63 ± 3	If CV ≤ 70%: scored 1	1
	(3) Sp-DCP	52 ± 4		1
	(4) Sp-DCP + γCD	47 ± 4		1
(II) 0.05%	(1) Blank Sp-DCP	87 ± 6	If CV >70%: scored 1If CV ≤ 70%: scored 2	1
	(2) Blank Sp-DCP + γCD	85 ± 4		1
	(3) Sp-DCP	83 ± 4		1
	(4) Sp-DCP + γCD	81 ± 2		1
	Total score (I and II)		(1) Blank Sp-DCP	2
			(2) Blank Sp-DCP+γCD	2
			(3) Sp-DCP	2
			(4) Sp-DCP+γCD	2

**Table 6 pharmaceutics-14-01147-t006:** Total FOS content (%) of FOS-loaded niosomal preparations and FOS/γCD complex storage at 4 °C, 30 ± 2 °C (75 ± 5% RH) and 40 ± 2 °C (75 ± 5% RH) for 0, 1, 3 and 6 months (*n* = 3, mean ± SD) The % FOS content was calculated based on 100% as initial drug content at 0 month.

Time (Month)	Formulations		
	Sp-DCP	Sp-DCP+γCD	FOS/γCD Complex
*5 ± 3 °C*			
1 Month	97.95 ± 0.70	98.44 ± 0.64	81.09 ± 0.92
3 Months	93.72 ± 0.73	95.21 ± 0.39	73.84 ± 0.68
6 Months	88.33 ± 0.54	92.75 ± 0.83	51.10 ± 1.18
*30 ± 2 °C (65 ± 5% RH)*			
1 Month	93.32 ± 0.53	95.13 ± 0.86	28.72 ± 0.30
3 Months	83.40 ± 0.78	87.37 ± 0.57	20.94 ± 0.73
6 Months	17.17 ± 0.59	23.67 ± 0.57	8.49 ± 0.70
*40 ± 2 °C (75 ± 5% RH)*			
1 Month	46.09 ± 0.88	56.34 ± 0.82	19.95 ± 0.60
3 Months	27.88 ± 0.71	36.70 ± 1.08	12.26 ± 0.36
6 Months	7.75 ± 0.83	10.68 ± 1.06	3.59 ± 0.70

## Data Availability

Not applicable.

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
