# Peer review of "Physicochemical and Stability Evaluation of Topical Niosomal Encapsulating Fosinopril/γ-Cyclodextrin Complex for Ocular Delivery"

_pharmaceutics, 2022, doi:10.3390/pharmaceutics14061147_

Round 1

Reviewer 1 Report

Hnin et al. have developed a niosomal formulation encapsulating fosinopril. The authors have evaluated the effect of addition of cyclodextrins and surfactants on the stability of the formulations. While there have been numerous instances of particle- based approaches employed for ocular application, this study is a good attempt at understanding which formulation parameters affect physicochemical stability of the construct. However a few comments must be addressed before the manuscript is deemed suitable for publication.

  1. Authors mention that the viscosity of niosomal samples -s ~1-2 mPa.s ( Lines 243-244). While the risk of blurring the vision would not occur, the reviewer is concerned about the possibility of quicker drainage that occurs with less viscous solutions. With such viscosities, retention in eye might be a challenge, which would prompt repeat dosing and poor patient compliance. Authors should consider and state the effect of low viscosity on efficacy of their proposed formulation.
  2. Authors describe the size/ zeta potential of their niosomal formulations in lines 258-263. Bimodal and trimodal distributions with pdi ~ 0.3-0.5 are not ideal and for all practical purposes are considered non-monodispersed. Is this data based on intensity mean? Authors should consider reporting number mean as that would be an accurate description of what is happening in the solution. Further, such polydispersed solutions will pose challenges in manufacturing as well defining a control strategy and authors must address this in the section.
  3. Encapsulation efficiencies of FOS seem to be on the lower end (< 40%). Authors should state the reason for the same. Was there an effort to try and increase the encapsulation?
  4. Figure 2 describes the release profile of the niosomal formulations. In 24 h, ~25% FOS is released. Again the reviewer would like to bring the first point up. With low viscosities, low encapsulation efficiency and a slow release, these formulations are at a risk of sub-optimal dosing. Authors should try and define and explain their strategy holistically. In theory ~25% release might work for certain cases but may not work in this case due to practical considerations.
  5. In section 3.5 the authors describe physicochemical aspects upon exposure to stability. The reviewer suggests moving stability indicating parameter, in this case the PDI,  to the main text and include that in Table 6. While encapsulation efficiency is definitely a criterion, particle based formulations are subject to more stringent criteria on polydispersity and must be included in the discussion. Again, the reviewer asks the author to consider practical implications of PDI. Most of the samples described in the study with PDI values >0.3 at t=0, will be considered "not-feasible" from a product perspective.

In the light of above comments, a major revision with additional clarification/ analysis or repeat experiments in certain cases is warranted.

Reviewer 2 Report

The authors developed a new niosomal eye drop containing fosinopril 16 (FOS) for lowering IOP of glaucoma patients. However, prostaglandin analogues and beta-blockers are the gold standard of glaucoma therapies currently. It needs to be determined if angiotensin-converting enzyme inhibitors have better efficacy and what it can add to the current practice. The authors need to work on the meaning of this study more. A few other comments are listed below:

  1. Introduction: Please also review and add more recent references. (Line 35 and Line 42)
  2. Result: The purpose of this eye drop is to control IOP, however the IOP data before and after experiment were not presented here.

Reviewer 3 Report

The authors developed niosomes for the ocular delivery of fosinopril.  The manuscript is well structured and the work in interesting. The developed niosomes were physiochemically characterized using adequate methodologies, their ex vivo permeation ability was studied as their in vitro biocompatibility in rabbit fibroblasts. However, major revisions are required to improve the manuscript before publication. Below the authors can find some suggestions and questions.

Keywords: please avoid repeating words from the title, such as γ-cyclodextrin; ocular delivery; fosinopril

Line 96: plese add Hydration medium compostion

Table 1 is very difficult to understand. Please improve it (e.g. give an ID to the different formulations, etc)

Lines 128-135: Please add missing details, such as dispersion medium (viscosity, refractive index, absorbance values), niosomes concentration

Line 197: please include concentration range for FOS

Line 201: 1x105, 5 must be in superscript letter

Line 223-231: Stability studies in simulated physiological conditions (as the authors did for the release experiments, PBS, pH 7.4, ocular temperature) would provide valuable information.

Table 3 and lines 262-262: the prepared nanoformulations have high PDI values. I do not agree whith the authors conclusion that the formulations are monodisperse. PDI values below 0.3 are usually considered to be acceptable as indicated in reference 31. This reference 31 refers to PDI values of 0.7 as an indicator of a very broad particle size distribution and not that PDI values of 0.7 are considered good.

Table 3: overall, the obtained EE are very low. Can the authors comment on that?

Line 338: the authors state that they excluded Brij® 76-niosomes from further studies not only due to particle aggregation but also because these present lower EE values. However, EE values for Span® 60-Niosomes are within the same range. Please correct.

Figure 1: the sizes of the formulations (a) and (b) displayed in the (images near 500 nm) are very different from the sizes determined by DLS (table 3, 245 for a and 262 for b), which suggests (as the high PDI values) that the samples are very polydisperse.

Figure 2: A sample with free FOS as control should be included in the release experiments to ensure the feasibility of the experiment and to assess that the free TO has no affinity to the cellophane membrane and that sink conditions are ensured.  Furthermore, the authors could elaborate on their choice of the duration for the release experiments (24h), since at this point only about 15-25% (depending on the formulation) of the drug was release. The authors could extend the release experiments to achieve total release.

Figure 3: I suggest ordering the concentration axis from lower to higher values.

Line 518: the authors state that the formulations exhibited physical and chemical stability for at least six months at 4°C.  However, as shown in table S2, size and PDI values significantly increased after 3 months for both studied niosomal formulaitons. Please correct.

Round 2

Reviewer 1 Report

Authors have provided clarifications and have significantly improved the manuscript. The manuscript can be deemed suitable for publication following one minor comment to be addressed:

1. Authors state the following in response to one of the reviewer's comments

"We have measured the drug content in the niosomal formulations according to ICH compared to the drug/CD complexes"

Authors should explicitly state that while particle size/ size distributions are stability indicating for particle based formulations, %EE is considered as a stability indicating parameter for this study in direct comparison to non-particulate counterparts. This will clarify the context in which these assessments have been made.

Author Response

1. Authors state the following in response to one of the reviewer's comments

"We have measured the drug content in the niosomal formulations according to ICH compared to the drug/CD complexes"

Authors should explicitly state that while particle size/ size distributions are stability indicating for particle based formulations, %EE is considered as a stability indicating parameter for this study in direct comparison to non-particulate counterparts. This will clarify the context in which these assessments have been made.

Response: Thank you for your comment and suggestion. We have added:

Lines 498-500: While particle size and size distributions are stability indicating for particle-based formulations, %EE is considered as a stability indicating parameter for this study in direct comparison to non-particulate counterparts.

Best regards,

Phatsawee Jansook

Reviewer 3 Report

The authors addressed all my comments and publication is deserved

Author Response

Thank you very much.

Sincerely,

Phatsawee Jansook